# Avoiding 3D Obstacles in Mixed Reality: Does It Differ from Negotiating Real Obstacles?

**DOI:** 10.3390/s20041095

**Published:** 2020-02-17

**Authors:** Bert Coolen, Peter J. Beek, Daphne J. Geerse, Melvyn Roerdink

**Affiliations:** Department of Human Movement Sciences, Faculty of Behavioural and Movement Sciences, Vrije Universiteit Amsterdam, Amsterdam Movement Sciences, Van der Boechorststraat 7, 1081 BT Amsterdam, The Netherlands; p.j.beek@vu.nl (P.J.B.); d.j.geerse@vu.nl (D.J.G.); m.roerdink@vu.nl (M.R.)

**Keywords:** HoloLens, mixed-reality headset, mixed-reality video feedback, walking adaptability, obstacle avoidance

## Abstract

Mixed-reality technologies are evolving rapidly, allowing for gradually more realistic interaction with digital content while moving freely in real-world environments. In this study, we examined the suitability of the Microsoft HoloLens mixed-reality headset for creating locomotor interactions in real-world environments enriched with 3D holographic obstacles. In Experiment 1, we compared the obstacle-avoidance maneuvers of 12 participants stepping over either real or holographic obstacles of different heights and depths. Participants’ avoidance maneuvers were recorded with three spatially and temporally integrated Kinect v2 sensors. Similar to real obstacles, holographic obstacles elicited obstacle-avoidance maneuvers that scaled with obstacle dimensions. However, with holographic obstacles, some participants showed dissimilar trail or lead foot obstacle-avoidance maneuvers compared to real obstacles: they either consistently failed to raise their trail foot or crossed the obstacle with extreme lead-foot margins. In Experiment 2, we examined the efficacy of mixed-reality video feedback in altering such dissimilar avoidance maneuvers. Participants quickly adjusted their trail-foot crossing height and gradually lowered extreme lead-foot crossing heights in the course of mixed-reality video feedback trials, and these improvements were largely retained in subsequent trials without feedback. Participant-specific differences in real and holographic obstacle avoidance notwithstanding, the present results suggest that 3D holographic obstacles supplemented with mixed-reality video feedback may be used for studying and perhaps also training 3D obstacle avoidance.

## 1. Introduction

In 1901, L. Frank Baum, the celebrated author of “The Wonderful Wizard of Oz”, alluded to something that we would now call augmented reality or mixed reality. In his novel “The Master Key” [1], a teenage boy was rewarded by the Demon of Electricity with a unique pair of spectacles, coined the “Character Marker”, which could project a key onto a person’s forehead indicating their character. What L. Frank Baum’s readers might have considered impossible at the time is now close to reality. Since the 1950s, remarkable progress has been made in augmenting the real world with layers of digital content: the initial bulky setups have been supplanted by mobile and wearable applications [2,3,4,5].

One promising development in the area of wearable applications is the Microsoft HoloLens mixed-reality headset (Microsoft Corporation, Redmond, WA, USA), which was released in March 2016 as a Developers Kit. The HoloLens is an untethered and optical see-through headset with a holographic display unit through which 3D holograms are not only overlaid, but also anchored to and interacting with the wearer’s environment. To ‘understand’ the real world in real time, the HoloLens is a ‘holographic’ computer equipped with an inertial measurement unit or IMU, four ‘environment-understanding’ cameras, a depth camera (Kinect v3 sensor), and mixed-reality capture. The HoloLens uses a set of algorithms collectively called Simultaneous Localization and Mapping (SLAM) to compute the position and orientation of the headset with respect to its surrounding, while at the same time mapping the structure of that environment [6], which is a feature also used nowadays in driverless cars, drones, Mars rover navigation, and even inside the human body [6,7,8,9,10,11]. This enables augmentation of the real world with location-aware 3D holographic digital content while moving about freely, potentially even in open environments.

The merging of real and digital worlds is referred to as mixed reality. It is an important development, allowing for progressively more natural interaction with both real and digital content. Gibson ([12], p. 1) pointed out that “when we explore the real world, we look around, walk up to something interesting, move around it, and go from one vista to another”. This is also possible in mixed reality, where real physical and digital objects co-exist in space and interact in real time. There is, in that regard, an important difference to note between mixed reality and the more commonly known virtual-reality headsets. In virtual reality, there is full control over the displayed virtual world, because users are visually disconnected from their physical surroundings. Virtual-reality headsets have proven useful in training, simulation, education, and gaming, but they preclude natural interaction with the real world. A case in point is walking with a virtual-reality headset [13]: there are infinitely large and diverse virtual environments to visit, but walking is strongly confined either to a limited physical space or to walking in place on some sort of stepping device such as an omnidirectional treadmill. The main challenge when using virtual-reality headsets while walking is that there is no direct connection between physical and virtual worlds. With mixed reality, this drawback has been resolved as visual perception of the real world and physical movements in the real world are unconstrained. The HoloLens provides the wearer with a natural view of the real world with its own full resolution [14], and digital information can be positioned anywhere in this world.

Although the HoloLens represents a promising emerging technology, with early-adopter scientists sharing their initial insights regarding its potential [15,16,17,18,19,20,21,22,23,24,25,26], the HoloLens still faces some challenges [16,25,26,27,28]. For one, there are technical limitations of the display such as a restricted augmentable field of view (FOV). Although the HoloLens does not constrain the peripheral vision of the real world, the FOV in which holograms are visible is much smaller and estimated to be about 30° wide and 17.5° high. Second, there are perceptual limitations such as the mismatch between focal and real distance (the HoloLens display has a fixed focal length of approximately 2.0 m). This may result in differences in visuomotor control between real and augmented environments [25]. Third, there are technical challenges related to tracking and spatial mapping, such as stable and accurate holograms at large distances, handling seamless occlusions between real and digital contents and the potential loss of tracking, predominately during fast turns, when the device cannot readily locate itself with respect to the world. Finally, the HoloLens is a Developers’ version and with a weight of 579 g, it is still relatively heavy. 

The objective of this study was to examine, given the aforementioned challenges, the potential use of HoloLens mixed reality for creating locomotor interactions in real-world environments enriched with holographic obstacles. Specifically, in Experiment 1, we used real and holographic obstacles of various but matched heights and depths to compare real to holographic obstacle-avoidance maneuvers. We expected participants to scale their crossing maneuvers to changes in obstacle dimensions for both obstacle conditions. We further expected the crossing strategy for holographic obstacles to be (1) more conservative (slower speed, greater margins) and/or (2) less successful (more collisions). The conservative crossing strategy expectation followed from [29], in which augmented-reality obstacle avoidance was compared to real obstacle avoidance. This particular setup involved 2D projection onto the ground plane with position-dependent perspective correction and stereoscopic goggles for a 3D obstacle percept. A slower approach speed and greater obstacle-crossing margins were found for the augmented-reality obstacles, suggesting that participants were more cautious when approaching these obstacles compared to real obstacles. The less successful crossing expectation was based on the relatively small augmentable FOV of the holographic display unit, which could preclude a full display of large holographic obstacles in close proximity. As the pickup of visual information about the obstacle is important both prior to and during obstacle negotiation [30], this could diminish avoidance success, especially for the trailing limb and with larger obstacle dimensions. In follow-up Experiment 2, we examined the efficacy of mixed-reality video feedback in altering deviating holographic obstacle-avoidance maneuvers observed in Experiment 1 that were dissimilar from real obstacle-avoidance maneuvers in terms of extreme margins and collisions.

## 2. Experiment 1: Comparing Holographic to Real Obstacle Avoidance

### 2.1. Materials and Methods

#### 2.1.1. Participants

A group of 12 healthy participants took part in this experiment. Participants were heterogeneous in gender (7 males, 5 females), age (mean [range]: 39 (21–64) years of age, with six participants within the range of 21 to 29 years of age and the other six ranging from 49 to 64 years of age), height (176 (155–195) cm, with nine participants being between 173 and 185 cm tall) and recruitment population (4 students, 3 colleagues, and 5 participants from outside the academic community). This heterogeneity allowed for an across-the-board assessment of the main research question. Participants did not have any medical condition that would influence walking or normal vision. Six participants were wearing contact lenses, and two used reading glasses in daily life, but not during the experiment. One participant had previous experience with using the HoloLens. 

#### 2.1.2. Ethics Statement

The experiments were approved by The Scientific and Ethical Review Board (VCWE) of the Faculty of Behavioural and Movement Sciences of the Vrije Universiteit Amsterdam (VCWE-2018-019). All participants provided written informed consent prior to participation. 

#### 2.1.3. Experimental Setup and Design

Experiment 1 involved stepping over real or holographic obstacles placed within a 2.1 m walking path. A start and stop box marked the beginning and end of the walking path (Figure 1A). The obstacle distance from the start box was always 1.2 m. The experiment comprised 2 blocks of 60 trials each, in which participants stepped either over real or holographic obstacles, with different obstacle height (0.0 m, 0.1 m, 0.2 m, 0.3 m, 0.4 m) and obstacle depth (0.02 m, 0.30 m) conditions (Figure 1A,B,C). Each condition was repeated six times. The order of the real and holographic obstacle-avoidance blocks was counterbalanced over participants in that six participants started with real obstacles (without wearing the HoloLens) and the other six started with holographic obstacles. Depth conditions were also counterbalanced over participants, while height conditions were block-randomized within and between participants. Each trial began by assuming a standing posture in the designated start box. Participants were instructed to walk naturally at a comfortable pace from the start box to the stop box while avoiding the real or holographic obstacle. An auditory tone signaled to start walking. Participants were instructed to finish the trial by standing with both feet in the stop box.

The real obstacles consisted of one or two horizontal hurdles made of aluminum bars held with magnets that could be fastened at different heights, as shown in Figure 1A. The bars dropped when touched. Holographic obstacles were presented with a HoloLens (Figure 2A; Developer version 1.0: https://www.windowscentral.com/hololens-hardware-specs) and consisted of one or two holographic hurdles, with the colors red and green (Figure 1B,C). A custom-made HoloLens application was built using the Unity3D 2017.3 engine for presenting these obstacles. The frame rate was 60 frames per second to ensure that the holograms and the matching coordinate system of the headset were in sync. The resolution per eye was set to 1268 × 720 pixels. Since the relatively small augmentable FOV of the HoloLens may preclude the full display of large holographic obstacles, holographic hurdles were preferred over solid 3D holographic obstacles to improve the interpretability of the obstacle’s dimensions and its location with respect to the ground. It is important to note that the holographic obstacles were placed using a spatial anchor, i.e., a local little map of the environment, to ensure that a hologram remained exactly at a specific spot in the real world. Furthermore, spatial anchors were persistent over time, which ensured that holographic obstacles were presented at the same location across participants and conditions. Since the HoloLens display has a fixed focal length of approximately 2.0 m, we set the distance between the first hurdle and the start box (1.2 m) such that the holographic hurdle was optimally viewed at a distance of about 2.0 m (assuming a HoloLens height of 1.75 m and an average obstacle height of 0.2 m). Moreover, at this distance from the start box, the holographic obstacle was fully in view, with a stable location on the floor. Before starting with the holographic obstacles block of trials, the pupillary distance was determined with the ‘HoloLens calibration’ app, while basic hand gestures were learned with the app ‘Learn gestures’ to enable interaction in mixed reality. Thereafter, participants played the game ‘Roboraid’ for at least 10 minutes to get used to the HoloLens. Finally, a holographic obstacle was projected together with a real obstacle of equal height and depth (Figure 2B) to check whether the holographic obstacle was perceived at the correct location, as set with the spatial anchor. All participants reported a visual overlap of real and holographic obstacles. The total duration of wearing the HoloLens before starting the measurements was approximately 30 minutes.

Participants’ movements were recorded with a setup consisting of three spatially and temporally integrated Kinect v2 sensors for the markerless registration of 3D full-body kinematics [31,32,33], which was recently validated for walking [34,35,36]. The Microsoft Kinect v2 sensor is a RGB-D camera, and the three sensors were positioned as depicted in Figure 1A. Appendix A show first-person views of holographic obstacle-avoidance trials, as seen through and recorded with the HoloLens.

#### 2.1.4. Data Pre-Processing and Analyses

The Kinect for Windows Development Kit (SDK 2.0, www.microsoft.com) provides 3D positions of 25 body points, including the spine, shoulder, ankle, and foot body points, at a sample rate of 30 Hz. To obtain finer-grained data of foot clearance, we also analyzed the depth images (Figure 3B) of the three Kinect v2 sensors. To this end, the depth image was first pre-processed for each Kinect sensor, and depth points belonging to the human body were mapped to 3D points (x, y, z), resulting in a 3D point cloud (Figure 3B). Subsequently, we extracted the 3D point clouds of each foot and determined the lower edges of the foot, which were sent to and stored on the main computer. These data are available at https://doi.org/10.5281/zenodo.3581238. 

Additionally, all data, including the depth data, of Kinect sensor 3, located at the front on the right side, were logged using Microsoft Kinect Studio (Figure 3A). A point of attention was the correct labeling of the left and right foot during occlusion episodes. Therefore, in the offline analysis, the already determined contours of each foot for Kinect sensor 3 were first inspected visually using the associated depth data, and, if necessary, labeling of the left and right foot was corrected or classified as missing in case of lack of visibility of the lower part of a foot, and these were considered ground truth in further analyses. In case of missing data of Kinect sensor 3, already determined point clouds from other Kinect sensors were used. In case of occlusion by the real obstacle, the point clouds of the Kinect sensor positioned at the side just in front of the obstacle were used (see https://doi.org/10.5281/zenodo.3581238).

The following kinematic parameters were calculated offline in MATLAB R2018a: maximum step height, crossing height, and clearance, all separately for the lead and trail foot, and mean forward velocity. Maximum step height (in m) was defined as the peak in the minimum vertical values of a foot’s point cloud anywhere between 0.8 and 1.6 m along the walking path. Crossing height (in m) was defined as the minimum vertical distance between the foot and the ground above hurdle 1 (at 1.2 m) and, if present, above hurdle 2 (at 1.5 m). To obtain a single value for the crossing height of a moving point cloud above the hurdle(s), the anterior–posterior lower edge of a foot’s point cloud was divided into 30 parts of equal length. Subsequently, the movement trajectories of each of these 30 parts were interpolated, and the minimal vertical distance of each part was taken above the hurdle(s). The crossing height was defined as the minimum of the so-obtained 30 minimal vertical distance values. By subtracting the obstacle height from crossing height at hurdle 1 (and 2 if present), we determined the foot clearance (in m). A negative foot clearance was considered indicative of a collision for holographic obstacles (for real obstacles, we simply documented how often the bar(s) fell). The mean forward velocity of the participant (in m/s) was defined as the distance traveled along the walking path between 0.1 and 1.8 m divided by time, using the spine shoulder data provided by the Kinect SDK. Three trials were excluded from the analysis because the recordings were stopped before the participant passed the obstacle (i.e., experimentation failure, all physical obstacles, one for participant 9 (dimension: 0.30 × 0.02 m) and two for participant 10 (dimensions: 0.20 × 0.02 m and 0.30 × 0.30 m), while five trials of the highest holographic obstacle (two for 0.02 m and 3 for 0.30 m depth) of participant 10 were excluded because the maximum step height of the trailing foot could not be determined because of missing values.

The structure of the dataset of outcome measures will be visualized with a 2D multidimensional scaling (MDS) plot. With MDS, a map of the relative positions of a number of objects is created, based on a table of inter-object distances [37]. Similar objects will appear in close proximity of each other while dissimilar objects appear farther apart. In that regard, MDS provides a means of pre-sorting or anomaly detection (i.e., dissimilar crossing techniques for real and holographic obstacles). Prior to the statistical analysis, we therefore created an MDS representation of the effect of obstacle type for each participant. Hereby, an obstacle-type object (real or holographic) of a participant was represented by a vector of all associated step heights and all foot clearances at hurdle 1 (and 2 if present) for both the lead and the trail foot, so each vector consisted of 250 elements (hurdle 1: 5 heights × 2 depths × 5 trials (2–6) × 4 outcome measures; hurdle 2: 5 heights × 5 trials (2–6) × 2 outcome measures). Then, the Euclidian distance was calculated between each pair of obstacle-type objects, a.k.a. the distance between the associated vectors. Subsequently, the calculated distance matrix (size 24 × 24 elements) with the distances between each pair of obstacle-type objects served as the input for MDS. 

#### 2.1.5. Statistical Analysis

We conducted an obstacle-type (real, holographic) by height (0.0 m, 0.1 m, 0.2 m, 0.3 m, 0.4 m) by depth (0.02 m, 0.30 m) repeated-measures ANOVA (alpha of 0.05) for maximum step heights, forward velocity, and for crossing heights and foot clearances above hurdle 1. Likewise, an obstacle-type (real, holographic) by height (0.0 m, 0.1 m, 0.2 m, 0.3 m, 0.4 m) repeated-measures ANOVA was conducted for crossing heights and foot clearances above hurdle 2. The first trial of the six repetitions was excluded from the analysis to prevent the possible influence of potential first-trial effects. The input for the statistical analyses was the median of the available values over repetitions 2 to 6. The effect sizes are reported in the form of *η_p_^2^*-values.

To ensure that the repeated-measures ANOVA would be validly applied, the assumption of normality was first tested using the Shapiro–Wilk test for normality; the vast majority (98%) of dependent variables were normally distributed (*p* > 0.05). The assumption of sphericity was verified according to Girden [38]. The Huynh–Feldt correction was applied if the Greenhouse–Geisser’s epsilon exceeded 0.75; otherwise, the Greenhouse–Geisser correction was used. Post-hoc pair-wise comparisons with Bonferroni adjustment were performed for significant main effects of height and for significant interactions.

### 2.2. Results

#### 2.2.1. Multidimensional Scaling: Visualizing the Effect of Obstacle Type

An MDS representation of the effect of obstacle type for each participant is visualized in Figure 4. All distances among real-obstacle conditions were short, indicating similarity in real-obstacle negotiation for all participants. Likewise, distances between real and holographic obstacle conditions were short for participants 2, 3, 5, 6, 9, and 12, indicating similarity in avoiding real and holographic obstacles for these participants. For participants 1, 8, and 11, in contrast, distances were considerably larger, indicating a dissimilarity between real and holographic obstacle crossing. As can be seen in Figure 5D, these three participants were consistently not raising their trail limb during holographic obstacle avoidance, resulting in a trail foot collision with the holographic obstacle (Figure 5B). In addition, distances were also larger for participants 4, 7, and 10, again indicating dissimilarity between real and holographic obstacle avoidance. As can be appreciated from Figure 5C, these three participants were consistently raising their lead limb exceptionally high during holographic obstacle avoidance, without any obstacle hits (Figure 5A).

#### 2.2.2. Collisions 

For real obstacles, we observed just a single collision for the trail foot at hurdle 2 with 0.4 m height. For holographic obstacles, substantially more collisions were observed. In Figure 5A,B, the distribution of collisions with the lead and trail foot at holographic hurdle 1 and 2 is depicted. A total of 39 (6.5% of the trials) and 32 (10.7%) lead-foot collisions were observed for holographic hurdles 1 and 2, respectively, which were for the most part caused by participant 11 (32 and 18 collisions, respectively; Figure 5A). For the trail foot, a total of 143 (21.7%) and 84 (28%) collisions were observed with holographic hurdles 1 and 2, respectively, mainly resulting from participants 1, 8, and 11, who always touched the holographic hurdle(s) (60 collisions per participant; Figure 5B). The remaining 21 lead and 47 trail-foot collisions mainly occurred for higher obstacles (0 m: 0, 0.1 m: 8, 0.2 m: 14, 0.3 m: 24, 0.4 m: 22). Since participants 1, 8, and 11 were also identified in the MDS representation as participants with a dissimilar holographic obstacle-avoidance maneuver, we decided to exclude them from further statistical analyses of trail foot outcomes because, in contrast to the other nine participants, they did not adjust their trail foot to holographic obstacles with heights 0.1 to 0.4 m (see also Figure 5D). 

#### 2.2.3. Maximum Step Height

The maximum step heights are depicted in Figure 6A,B, which increased with obstacle height as evidenced by main effects of obstacle height for both the lead foot (*F*(1.703,19.057) = 309.308, *p* < 0.001, *η_p_^2^* = 0.966) and trail foot (*F*(2.029,16.223) = 217.237, *p* < 0.001, *η_p_^2^* = 0.964). Post-hoc pairwise comparisons revealed significant differences between all obstacle heights (*p* < 0.001). The maximum step height further increased with deeper obstacles, as evidenced by the main effects of obstacle depth for both the lead foot (*F*(1,11) = 51.114, *p* < 0.001, *η_p_^2^* = 0.823) and trail foot (*F*(1,8) = 156.131, *p* < 0.001, *η_p_^2^* = 0.951). Finally, there was a significant obstacle height × depth interaction for the trail foot (*F*(4,32) = 3.552, *p* = 0.017, *η_p_^2^* = 0.307), indicating that the increase in maximum step height with deeper obstacles was only significant for 0.1 m, 0.2 m, and 0.3 m obstacle heights. 

The obstacle type had no effect on maximum step height for the trail foot (*F*(1,8) = 0.770, *p =* 0.746, *η_p_^2^* = 0.014) and only a small effect for the lead foot (*F*(1,11) = 5.134, *p =* 0.045, *η_p_^2^* = 0.318; holographic obstacles: 0.436 m, real obstacles: 0.359 m). However, as can be appreciated from Figure 6A,B, the between-subject variability was about three times higher for holographic than real obstacles, suggesting that participants varied widely in maximum step heights. Figure 5C,D shows that this was indeed the case. Here, the differences in maximum step heights between holographic and real obstacles are depicted. Some participants raised their leg higher to step over a real obstacle (negative values), while others raised their leg higher over a holographic obstacle (positive values), with participants 4, 7, and 10 raising their lead foot extremely high to step over 0.1 m, 0.2 m, 0.3 m and 0.4 m holographic obstacles (i.e., much higher than the upper bound of the 95% confidence interval of the difference in maximum step heights between holographic and real obstacles). This observation was corroborated by the greater distances for these three participants in the MDS representation (Figure 4). Note that if we exclude participants 4, 7, and 10 for the lead foot maximum step height, the significant main effect of obstacle type disappears (*F*(1,8) = 0.806, *p =* 0.398, *η_p_^2^* = 0.092); then, the maximum step heights for holographic and real obstacles were similar (holographic obstacles: 0.338 m, real obstacles: 0.337 m). 

#### 2.2.4. Crossing Height and Clearance above Hurdle 1 

The crossing height above hurdle 1 increased with obstacle height for both the lead foot (*F*(1.929,21.220) = 338.845, *p* < 0.001, *η_p_^2^* = 0.969; Figure 6C) and trail foot (*F*(4,32) = 244.587, *p* < 0.001, *η_p_^2^* = 0.968; Figure 6D). Post-hoc pairwise comparisons revealed significant differences between all obstacle heights. There were no significant main effects of obstacle type and obstacle depth, nor any significant interactions. 

The vertical foot–obstacle distance (foot clearance) above hurdle 1 is also shown in Figure 6C,D. Significant main effects were found for obstacle height (lead foot: *F*(4,44) = 14.468, *p* < 0.001, *η_p_^2^* = 0.568, trail foot: *F*(4,32) = 26.785, *p* < 0.001, *η_p_^2^* = 0.770). Pairwise comparisons revealed significant differences between 2D (0 m) and 3D (0.1 m, 0.2 m, 0.3 m, 0.4 m) obstacles, without significant differences in clearance among the four 3D obstacle heights. There were no significant main effects of obstacle type and obstacle depth, nor any significant interactions. 

#### 2.2.5. Crossing Height and Clearance above Hurdle 2

As for hurdle 1, the crossing height above hurdle 2 also increased with obstacle height (Figure 6E,F), for both the lead foot (*F*(1.757,19.323) = 263.370, *p* < 0.001, *η_p_^2^* = 0.960 and the trail foot (*F*(2.527,20.214) = 219.551, *p* < 0.001, *η_p_^2^* = 0.965); crossing heights differed significantly between all obstacle heights, except between 0.3 and 0.4 m for the trail foot. There were no significant main effects of obstacle type, nor any significant interactions. 

For foot clearance above hurdle 2, significant main effects of obstacle height were observed for both the lead foot (*F*(1.757,19.323) = 15.265, *p* < 0.001, *η_p_^2^* = 0.581; Figure 6E) and the trail foot (*F*(2.527,20.214) = 15.337, *p* < 0.001, *η_p_^2^* = 0.657; Figure 6F). In general, pairwise comparisons revealed significant differences between 2D (0 m) and 3D (0.1 m, 0.2 m, 0.3 m, 0.4 m) obstacles, with overall greater margins for 3D than 2D obstacles (except for the trail foot between 0 m and 0.4 m), without significant differences in clearance among the four 3D obstacle heights (except for a slight but significant greater margin for the 0.4 m obstacle compared to the 0.1 m obstacle for the lead foot). There were no significant main effects of obstacle type, nor any significant interactions.

#### 2.2.6. Mean Forward Velocity

The mean forward velocity decreased with obstacle height (*F*(4,44) = 38.672, *p* < 0.001, *η_p_^2^* = 0.779; with significant differences between all obstacle heights, *p* < 0.001), while forward velocity increased with obstacle depth (*F*(1,11) = 15.276, *p* = 0.002, *η_p_^2^* = 0.779; from 0.873 m/s for 0.02 m depth obstacles to 0.901 m/s for 0.3 m depth obstacles). The observed obstacle height × depth interaction (*F*(4,44) = 4.690, *p* = 0.003, *η_p_^2^* = 0.299) revealed no significant effect of depth on forward velocity for 2D obstacles and a smaller height-dependent decrease in forward velocity for the 0.3 m deep 3D obstacles than for the 0.02 m deep 3D obstacles. No further significant interactions were found. In addition, no significant difference was found between holographic and real obstacles (*F*(1,11) = 0.770, *p =* 0.399, *η_p_^2^* = 0.065; real obstacles: 0.894 m/s, holographic obstacles: 0.880 m/s). 

### 2.3. Discussion

The aim of Experiment 1 was to examine the potential of mixed reality for creating locomotor interactions by comparing obstacle-avoidance maneuvers for real and holographic obstacles of various matched heights and depths. Importantly, both obstacle-type conditions induced scaling to changes in obstacle heights and depths in terms of lead and trail foot maximum step heights (increased with obstacle height and depth), lead and trail foot crossing heights (increased with obstacle height), and forward velocity (decreased with obstacle height, increased with obstacle depth). Lead and trail foot clearances were overall smaller for 2D than 3D obstacles. We found no significant effects of obstacle type, except for a small effect for the lead foot (i.e., increased maximum step height for holographic obstacles), which could be attributed entirely to three participants with extreme lead-foot margins (Figure 5C, participants 4, 7, and 10, which were also identified with the MDS representation in Figure 4 as participants for which holographic-obstacle crossing was dissimilar from real-obstacle crossing). Margins for the lead foot of the remaining participants fell within the range reported in previous studies with similar obstacle dimensions; see for example [39].

The number of observed obstacle collisions were much higher for holographic than for real obstacles. In a research setting with full vision and normal lighting, collision rates of about 1–2% have been reported for young healthy participants [39,40], which were largely attributable to trail foot contacts (i.e., 67–100%, see [39,40,41,42,43,44]). In these studies, maximum obstacle height did generally not exceed approximately 0.3 m, while depths varied between 0.003 and 0.05 m, in any case with obstacle dimensions being much smaller than the 0.4 m high × 0.3 m deep maximal obstacle dimension used in our study. In addition, these studies typically examined obstacle avoidance during stable locomotion or discrete stepping tasks, whereas in our setup roughly two steps were taken before crossing the obstacle. If, for the sake of comparison, we would exclude the 0.3 m depth condition and only consider the 0.02 m depth condition, then the collision rates for holographic obstacles were not too dissimilar from the previously reported 1–2% in the literature. In fact, our participants did not show any collisions for real obstacles while for holographic obstacles (i.e., hurdle 1), the collision rate was on average 6.5% for the lead foot and 21.7% for the trail foot (but 5.1% after excluding participants 1, 8, and 11 who did not raise their trail foot (Figure 5B,D); these participants were also identified with the MDS representation as participants for which holographic-obstacle crossing was dissimilar from real-obstacle crossing; Figure 4).

In sum, for participants identified with MDS as having similar avoidance maneuvers for real and holographic obstacle crossings, holographic obstacle avoidance scaled with obstacle dimensions, with collision rates and margins roughly similar to previous reports. This positive finding notwithstanding, we also identified participants with dissimilar avoidance maneuvers between real and holographic obstacle negotiations. In particular, we found that three participants consistently did not raise their trail foot, while another three participants raised their lead foot extremely high. Remarkably, it seemed that these participants were not aware of their dissimilar holographic obstacle-avoidance maneuvers. Their dissimilar responses were likely not attributable to differences in body height, because all participants were well able to cross real obstacles, and dissimilar crossings already occurred at 3D holographic obstacles of 0.1 m height. Thus, body height had no obvious relation with dissimilar obstacle-crossing subgroups, albeit that the shortest two participants raised their lead leg extremely high. There are several other factors that may have promoted the observed dissimilar responses, including the viewing capabilities, lighting conditions, cognitive overload, cautiousness, shape, and fragility of the obstacles, but one major difference between holographic and real obstacles in the current setup is the lack of feedback on holographic obstacle-avoidance success. Participants may be unaware of hitting a holographic obstacle, such as for the three participants that were consistently not lifting their trail foot, or may adopt a more conservative crossing margin in order to reduce the risk of colliding with the holographic obstacle, such as that seen in the three participants with extreme lead-foot margins. In follow-up Experiment 2, we examined the efficacy of mixed-reality video feedback in altering holographic obstacle-crossing maneuvers in these two subgroups of participants with dissimilarity between real and holographic obstacle-avoidance maneuvers.

## 3. Experiment 2: Mixed-Reality Video Feedback for Altering Avoidance Maneuvers

The HoloLens already allows for mixed-reality video feedback using built-in mixed-reality capture for recording videos of real environments into which holographic information is blended. Considering the limited augmentable FOV of the HoloLens, which may restrict direct visual information about the interaction between the foot and the holographic obstacle, we expected that holographic obstacle avoidance could be improved by providing mixed-reality video feedback on obstacle collisions as well as various aspects of the performed avoidance maneuver itself (e.g., foot-obstacle distance, timing and fluency of the maneuver, overcompensations, foot positioning). To examine this expectation, we used a second HoloLens to generate a video of a side view of both real and holographic obstacle negotiation, which was shown to the participant at the end of each trial. Participants started with a block of trials without feedback (baseline trials), followed by a block of video-feedback trials to examine the time course of adjustments, and ending with a final block of trials without feedback to quantify the extent to which the adjustments, if any, were retained.

### 3.1. Materials and Methods

Experiment 2 was conducted directly after Experiment 1 with the same participants and experimental setup, and comprised two blocks of 30 trials each, in which participants stepped over real and holographic hurdles of always 0.3 m height × 0.02 m depth (see also Figure 2C,D), challenging obstacle dimensions considered suitable for studying the effect of mixed-reality video feedback. To visualize the obstacle-avoidance maneuver together with the (holographic) obstacle, a second HoloLens, providing built-in mixed-reality capture to record the real environment merged with 3D holograms as a video, captured obstacle-avoidance maneuvers from the side and streamed them to a remote computer using Windows Device Portal. Each real and holographic obstacle block started with two familiarization trials stepping over the obstacle to prevent the influence of potential first-trial effects. Next, five baseline trials without feedback were conducted, followed by 18 mixed-reality video-feedback trials (feedback video of previous trial directly before each trial), and finally five follow-up trials without feedback to examine whether the effects, if any, were retained in the absence of mixed-reality video feedback. The order of the real and holographic obstacle-avoidance blocks was counterbalanced over participants. Participants were instructed to walk naturally at a comfortable pace from the start box to the stop box while avoiding the real or the holographic obstacle. During the feedback trials, participants viewed the recorded (mixed-reality) video after each trial at least once or more often upon request. No further instructions or verbal feedback were given. Representative mixed-reality feedback videos are given in Appendix A. 

After the final trial, participants were asked to complete a purpose-designed questionnaire with items addressing user comfort, the realism of the presented holographic obstacles, and the usefulness of mixed-reality video feedback; participants were also asked to make suggestions for improvement of the HoloLens and the mixed-reality video feedback.

### 3.2. Results 

#### 3.2.1. Multidimensional Scaling: Visualizing the Effect of Feedback for Real and Holographic Obstacles 

In Figure 7, an MDS representation of the effect of mixed-reality video feedback is depicted for real and holographic obstacle-avoidance trials (trials 3 to 30, color-coded for baseline, feedback, and follow-up trials), derived from the clearance and maximum step height of both feet for the three subgroups of participants with either similar crossing maneuvers for real and holographic obstacles or dissimilar crossing maneuvers (i.e., extreme lead-foot margins or not raising the trail foot with holographic obstacles), as identified in Experiment 1. For all subgroups, the distances between all real obstacle crossings were small, indicating similarity for real obstacle-avoidance maneuvers among the baseline, feedback, and follow-up trials. However, for holographic obstacle-avoidance trials, there appeared to be a difference between the five baseline trials (trials 3 to 7), the feedback trials (trials 8 to 25), and the follow-up trials (trials 26 to 30), particularly so for participants 1, 8, and 11 without trail-foot adjustments (Panel E) and only slightly so for participants 4, 7, and 10 with extreme lead-foot crossing margins (Panel C), and not for the participants with similar crossing maneuvers for real and holographic obstacles (Panel A). Mixed-reality video feedback was quite powerful, in that trial 8 (the first trial after first receiving video feedback) already seemed to deviate somewhat from the cluster of baseline trials, suggesting that participants quickly adjusted dissimilar maneuvers (particularly the participants not raising their trail foot) in the course of mixed-reality feedback trials. This was substantiated by the distances between individual holographic obstacle trials and the average over all real obstacle trials (Panels B, D, and F), which quickly changed from trial 8 onwards for participants not raising their trail foot (Panel F). Furthermore, the MDS representation suggests that the effects induced by video feedback were largely retained in follow-up trials without feedback (trials 26–30), especially so for participants without trail-foot adjustments (Panels E and F).

#### 3.2.2. Crossing Heights: the Effect of Mixed-Reality Video Feedback

Figure 8 depicts crossing heights for the lead and trail foot at holographic hurdle 1. Overall, we confirmed the identification of participants with dissimilar crossing maneuvers between real and holographic obstacles of Experiment 1. That is, the baseline lead-foot crossing heights of participant 10, participant 7, and to a lesser extent participant 4 fell outside the blue band of Panel A (signaling a difference in lead-foot holographic obstacle crossing height between these three participants and the other nine participants) and outside the red band of Panel C (signaling a difference in lead-foot crossing height between real and holographic obstacles for these three participants; see also the single asterisks in Table 1). Lead-foot crossing heights were greater, which was consistent with the identified extreme lead-foot margins observed for participants 4, 7, and 10 in Experiment 1. Likewise, the three participants identified in Experiment 1 as not raising their trail foot (participants 1, 8, and 11) also showed baseline trail-foot crossing heights outside the blue band (signaling differences between these three participants and the other nine participants; Panel B) and the red band (signaling a difference between real and holographic obstacles for these three participants; Panel D; see also the single asterisks in Table 1). Trail-foot crossing heights were close to zero (and well below the obstacle height of 0.3 m, horizontal green lines in Figure 8), indicating that they did not raise their trail foot during the baseline trials of holographic obstacle crossing.

With mixed-reality video feedback, participants gradually reduced their extreme lead-foot crossing heights (Figure 8A,C) and quickly raised their trail foot to above the obstacle height of 0.3 m (Figure 8B,D). To examine individual effects of the mixed-reality feedback, we presented in Table 1 each individual’s 95% confidence intervals for the baseline trials, the feedback trials, and the follow-up trials. Double asterisks indicate confidence intervals deviating from those of the baseline trials. After receiving mixed-reality feedback, this was the case for participants 4 and 7: they both reduced their extreme margins considerably, albeit gradually, with confidence intervals deviating from the baseline trials in respectively the ninth and fifth block of five consecutive feedback trials (Table 1, Figure 8A,C) and for participants 1, 8, and 11: they all raised their trail foot already in the first block of five consecutive feedback trials (Table 1, Figure 8B and 8D). These effects were retained in the follow-up block of trials without feedback for participants 1, 4, and 8 (Table 1), with crossing heights close to those observed for real obstacles (red bands in Figure 8C and 8D). Thus, five of the six participants identified with dissimilar crossing maneuvers adjusted their maneuvers after receiving mixed-reality video feedback, and these effects were retained in three of these five participants in the absence of video feedback. An exception was participant 10, for whom no adjustments in lead-foot margins were observed after receiving mixed-reality video feedback; note that for real obstacles, this participant had the highest lead-foot obstacle crossings of all participants. Furthermore, the trail-foot modifying effect was smaller in magnitude for participant 11 and often not sufficient, resulting in collisions. Moreover, this small trail-foot modification effect was not retained in this participant (Table 1, Figure 8B,D). Note that crossing 0.3 and 0.4 high obstacles in Experiment 1 proved physically demanding for participant 11 (Figure 5C), who in the baseline trials of Experiment 2 also showed many lead-foot collisions with the holographic hurdle (light-blue lines in Figure 8A,C), which was presumably attributable to increased fatigue (as confirmed by the results of the questionnaire, see below). 

#### 3.2.3. Questionnaire: User Comfort, Realism of Holographic Obstacles, and Perceived Usefulness of the Mixed-Reality Video Feedback

We refer to Appendix A for the individual participant’s scores, as well as their comments on items related to user comfort, the realism of holographic obstacles, and the usefulness of mixed-reality video feedback. With regard to “user comfort” items (Questions 1.1 to 1.8 and related open question), seven out of 12 participants commented on the limited FOV of the holographic display, which negatively influenced holographic obstacle avoidance. However, participants indicated that the HoloLens itself did not block their view (cf. Questions 1.5 and 1.6). Participant 4 commented on the weight of the headset, while participant 11 commented on getting physically fatigued in the course of the experiments. Participants’ overall impression of the comfort of wearing the HoloLens device was neutral (cf. Questions 1.1, 1.7, and 1.8). Participants identified in Experiment 1 with extreme lead-foot margins (participants 4, 7, and 10) overall rated the user comfort somewhat lower than the other participants. That is, after transforming the scores of the three negatively phrased questions, their overall score over Questions 1.1 to 1.8 was 3.29 compared to 2.38 and 2.00 for the subgroups of participants with similar crossing maneuvers for real and holographic obstacles and participants not raising their trail foot, respectively. With regard to the “realism of the holographic obstacle” items (cf. Questions 2.1 to 2.13), Question 2.11 (“I had to lift my foot higher to step over holographic obstacles than over real obstacles”) proved quite informative. The overall score was neutral (2.58), but the three participants identified with extreme lead-foot margins strongly agreed with this question, with scores of 1 (participants 4 and 7) and 2 (participant 10). Participants rated the holographic obstacles as realistic (Question 2.13), with a score of 7.58 on a 10-point scale ranging from 1 (not realistic) to 10 (very realistic). With regard to “usefulness of mixed-reality feedback” items (Questions 3.1 to 3.5), participants seemed to agree that mixed-reality video feedback was useful in improving their holographic obstacle-avoidance maneuvers.

### 3.3. Discussion

In Experiment 2, we examined the efficacy of mixed-reality video feedback for altering dissimilar holographic obstacle-avoidance maneuvers. For all subgroups of participants, real obstacle crossings were similar across baseline, feedback, and follow-up trials, as evidenced by short distances in the MDS plot (Figure 7). For trials involving holographic obstacles, visual separations appeared between the block of baseline trials and the blocks of feedback and follow-up trials, particularly so for participants showing no trail-foot elevation in Experiment 1, suggesting that mixed-reality video feedback can be a fast (i.e., immediately after viewing feedback; Figure 8B,D) and also fairly effective way to bring about changes in holographic obstacle crossings (i.e., retention of the effect at follow-up trials without feedback), as confirmed with an evaluation of individual confidence intervals for baseline, feedback, and follow-up trials (Table 1). The questionnaire results shed light on the user comfort, the realism of the holographic obstacles, and the usefulness of the mixed-reality video feedback, which may aid in evaluating both technological and human-factor aspects that may or may not be associated with dissimilar holographic obstacle-avoidance maneuvers of specific participants, as will be discussed below. 

## 4. General Discussion

The overarching aim of the present study was to examine the potential of HoloLens mixed reality for creating realistic locomotor interactions. In Experiment 1, we compared avoidance maneuvers between crossing real and holographic obstacles that differed systematically in height and depth. Similar to real 3D obstacles, holographic 3D obstacles elicited avoidance maneuvers that scaled with obstacle dimensions. No significant effects of obstacle type were found for maximum step height, crossing height, or foot clearance, except for a slightly but systematically increased lead-foot step height for holographic obstacles, which proved attributable to three participants with extreme lead-foot elevation. Three other participants did not elevate their trail foot in response to holographic obstacles. In follow-up Experiment 2, we examined the efficacy of mixed-reality video feedback in altering such dissimilar holographic avoidance maneuvers, which proved to be fast and fairly effective in inducing changes in avoidance maneuvers, particularly so for the participants who did not raise their trail foot. In this General Discussion, we first focus on the (limitations in the) interplay between HoloLens technology (e.g., FOV) and human-factor aspects of the wearer (e.g., familiarization, fatigue, age, height) in relation to the holographic obstacle-avoidance effects found, and then we examine the opportunities for future mixed-reality applications for studying and training locomotor interactions. 

### 4.1. Limited Augmentable FOV Hampers Visual Information of the Obstacle during Obstacle Crossing

Having sufficient visual information about the obstacle is crucial for successful obstacle avoidance. This information can be used to plan the obstacle avoidance in advance (feedforward), as well as to modify avoidance behavior on-line. The participants in the present study had a (close to) optimal view of the holographic obstacle at the onset of the trial, during which the holographic obstacle was fully in view (Figure 1), thus providing sufficient visual information about the obstacle for planning an avoidance maneuver in advance. In contrast, on-line visual information about the obstacle and the foot–obstacle relation during the crossing maneuver was reduced due to the limited FOV of the HoloLens. In order for a participant to view the holographic obstacle while crossing, he or she had to make large unnatural head adjustments, which most participants refrained from doing (see videos of first-person views of holographic obstacle avoidance in Appendix A and the varied responses to Question 2.7 in Appendix A). Thus, most participants thus had to rely on feedforward visual information about the obstacle determined earlier in the approach phase, which may partly explain the increased collision rates with holographic obstacles as well as the increased lead-foot clearances with the holographic obstacles. Indeed, previous studies have shown that removing on-line foot–obstacle information (using goggles that blocked the lower visual field) resulted in an increased risk of obstacle contact and in some cases even increased foot clearances [40,45], which was attributed to a reliance on feedforward (rather than on-line) information about the obstacle. 

A restricted FOV may also affect the knowledge about the location and dimensions of the obstacle, which is known as obstacle memory [46] and could particularly impact upon trail-foot performance because neither the foot nor the obstacle are visible when the trail foot negotiates the obstacle. This finding emanated from a study [46] in which participants crossed a 3D obstacle for 25 trials after which the 3D obstacle was replaced with marks on the ground indicating its position. In the subsequent 25 trials, participants crossed the ‘3D obstacle’ by imagining it still to be there. Foot placement was unaffected, but failure rates increased up to 9% for the lead foot and up to 47% for the trail foot. Thus, when guided solely by obstacle-height memory, obstacle-avoidance success was reduced, particularly so for the trail foot. The authors concluded that viewing the obstacle during the approach phase facilitated obstacle memory, which particularly affected trail-foot obstacle-avoidance success [46].

Thus, the size of the augmentable FOV plays an important role in the pickup of obstacle information during holographic obstacle crossing. The FOV of about 30° wide and 17.5° high of the current HoloLens is limiting holographic obstacle avoidance in that regard, as large nearby obstacles do not fit within that FOV. However, that being said, the technology in question is evolving rapidly. For example, Magic Leap already introduced an untethered see-through headset with a FOV of 40° wide and 30° high, using the same waveguide display technology as HoloLens but closer to the wearer’s eye. Magic Leap partly blocks peripheral vision, and wearing glasses is not as comfortable under Magic Leap as under HoloLens. Microsoft introduced on 24 February 2019 the HoloLens 2 Developer Edition, with an increased augmentable FOV of 43° wide and 29° height. New features include eye and hand tracking, and HoloLens 2 is equipped with a Kinect for Azure depth sensor. Thus, the FOV limitation for holographic obstacle avoidance will at some point likely be solved by the mixed-reality technology push. 

### 4.2. Human-Factor Aspects May Influence Holographic Obstacle Avoidance

HoloLens wearers typically require a certain familiarization time to become accustomed to mixed-reality environments. Mostly due to the limited FOV, wearers move their head more and their eyes less compared to unconstrained visual exploration of the environment. Furthermore, (larger, nearby) holograms will clip within the smaller FOV than the overall visual field available. This is similar to the experience of wearing a pair of glasses for the first time, which also requires a period of familiarization to get used to the FOV set by the size of the glasses and the boundaries of the frame. We tried to familiarize participants with the HoloLens by playing the game “Roboraid” and by letting them wear the HoloLens for approximately 30 minutes before starting the measurements. We noticed that participants varied in the time required to get used to the HoloLens mixed reality. Participants were heterogeneous in age (ranging from 21 to 64 years) and background (four students, three staff members, and five participants from outside the academic community), with different affinity with new technologies, experimental protocols, and research settings (which all could contribute to cognitive overload). From this arguably somewhat small sample, the four students appeared to rapidly pick up the new technology and to actively explore the mixed-reality environments. Remarkably, none of the students was identified as a participant with a dissimilar crossing maneuver in Experiment 1. Other human-factor aspects, such as age, could not be readily linked to the subgroups of participants with dissimilar holographic obstacle avoidance maneuvers. For example, ages were seemingly equally distributed across the three subgroups (i.e., participants with similar crossing maneuvers were on average 33 years of age [but the oldest participant in this subgroup was 60 years of age], participants with extreme lead-foot margins were on average 46 years of age [but the youngest participant in this subgroup was 29 years of age], and participants without trail-foot elevation were on average 45 years of age [but the youngest participant in this subgroup was 22 years of age]). As already discussed in Section 2.3, differences in body height seemed also unrelated to participants with dissimilar obstacle crossing, albeit that the shortest two participants raised their lead leg extremely high. Finally, wearing the HoloLens for a prolonged time (maximally 50 minutes in Experiment 1) did not result in discomforts (see also the Appendix A). Physical fatigue, as developed by participant 11 in the course of the experiments, demonstrably affected obstacle-avoidance performance, yet this applied to both real and holographic obstacles. Future studies with a larger sample size could further zoom in on the effect of human-factor aspects on holographic obstacle avoidance.

### 4.3. Potential Benefits of Mixed-Reality Video Feedback for Holographic Obstacle Avoidance

Feedback is very powerful. After participants hit a real obstacle, a large increase in foot clearance in subsequent trials has been reported [44,47,48]. Feedback of collisions could also be related to enduring cautious avoidance maneuvers in studies on negotiating obstacles in augmented reality [29] and virtual reality [30]. Mixed-reality video feedback, as implemented in the current study through mixed-reality capture as a video (i.e., a built-in capability of the HoloLens), seems less sensitive to such overcompensations over longer periods of time because, apart from signaling collisions during obstacle negotiation, it further provides information about the quality of the obstacle-avoidance maneuver on a trial-to-trial basis, including information on the foot-obstacle distance, timing and fluency of the movement, foot positioning, and overcompensations. Participants simply looked at the video directly after each feedback trial, and trail-foot collisions as well as extreme lead-foot margins were rapidly and gradually corrected, respectively (Figure 8; Table 1; see also the Appendix A), without any instructions, remarks, or directions in that regard from the experimenters. Thus, mixed-reality video feedback, providing participants with both knowledge of results and knowledge of performance, played a key role in improving dissimilar holographic obstacle avoidance. However, this is not necessarily the ideal form of feedback. Other forms of feedback, such as direct haptic, auditory, and various forms of visual feedback on holographic obstacle collision could be explored in this context, as well as other environmental factors that can influence foot clearance, such as lighting conditions, general cautiousness, and the perceived fragility of obstacles [49].

### 4.4. A Look into the Future: Applications of Mixed-Reality Technology

In 1901, L. Frank Baum presented augmented reality as a concept of science fiction [1]. Today, augmented reality and mixed reality have become a reality [2,3,4,5], which has been fueled by the rapid advent of small and powerful mobile technologies, sharply dropping costs, and continuous progress in visual computing algorithms, 3D graphics capabilities, and depth sensing cameras. The HoloLens already carries a number of advanced features for mixed-reality applications. For example, all the tracking processes are performed on the headset (inside-out tracking), and rich data streams from several sensors can be collected, processed in real-time on the device, or sent over Wi-Fi to another computer to perform computationally more demanding calculations. Furthermore, digital information can be precisely anchored in the real world by means of spatial anchors. Emerging technologies such as Azure Spatial Anchors allow sharing such anchors across multiple devices in the cloud—for example, another HoloLens or smart phones (using frameworks as ARKit and ARCore)—and setting up an augmented-reality cloud, for example for an indoor navigation application.

With regard to locomotor interactions, mixed reality could offer new or enhanced opportunities for walking-adaptability training. In walking, it is important to be able to adapt to environmental circumstances, including the ability to avoid obstacles. Currently, walking-adaptability training is done with real obstacles or with 2D projections onto the walking surface [50,51,52,53,54,55]. Compared to 2D projections, the HoloLens offers the possibility to go beyond 2D by presenting 3D holographic obstacles. Compared to real obstacles, 3D holographic obstacles are safer and can be made to suddenly appear, which could prove beneficial for obstacle-avoidance training. Furthermore, holographic obstacles can be easily adapted and customized, and be part of an automatically administered protocol in which obstacle characteristics (timing, number, size) can be varied depending on the participant’s progress (see for a related example [55]). Moreover, holographic obstacles offer a mobile setup. We are currently working on a mixed-reality application coined ‘Holobstacle’, in which holographic content is controlled on the basis of features of a wearer’s movement and/or environment. In this application, a second person (e.g., experimenter, therapist) can see both the participant and the shared holographic content with a second HoloLens or smartphone. Moreover, the second person can control the shared holographic content. Subsequently, mixed-reality capture videos can be recorded from the point of view of the second person and presented to the participant to provide feedback about the obstacle-avoidance maneuvers made.

Mixed reality could also offer opportunities for locomotor interactions in other domains, such as sport or the military (note that Microsoft recently closed a deal to supply the US Army with a version of HoloLens 2 [56], enhanced with a Flir thermal camera). In general, the distinguishing opportunity offered by mixed reality is moving freely in the real world enriched with merged digital content. In order to promote free movement in mixed reality, future studies should investigate the quality and performance of the headset tracking, spatial mapping, and spatial anchors, extended to large-scale open environments.

## 5. Conclusions

Although mixed reality is still an emerging technology, 3D holographic obstacles, akin to real obstacles, elicited obstacle-avoidance maneuvers that scaled with obstacle dimensions. Some participants’ holographic obstacle-avoidance maneuvers were dissimilar from real obstacle-avoidance maneuvers, by either consistently not raising their trail foot or by crossing the obstacle with extreme margins. These dissimilar holographic obstacle-avoidance maneuvers were adjusted after receiving mixed-reality video feedback, and improvements were largely retained in a subsequent block of trials without feedback. Participant-specific differences between real and holographic obstacle avoidance notwithstanding, our study suggests that 3D holographic obstacles, supplemented with mixed-reality video feedback, has potential for studying and perhaps also training 3D obstacle avoidance. 

## Figures and Tables

**Figure 1 sensors-20-01095-f001:**
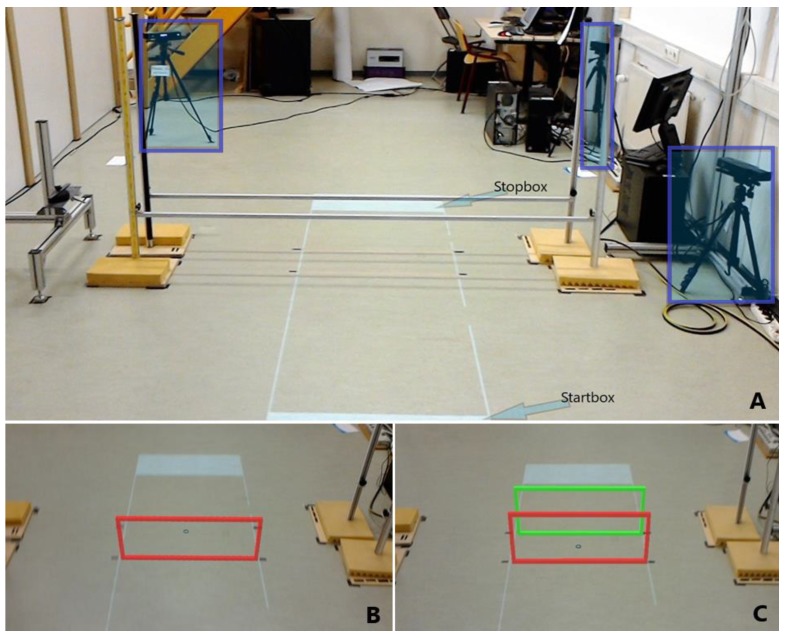
(**A**): Real obstacle with two hurdles (height 0.4 m, depth 0.3 m). White start and stop boxes projected onto the floor marked the beginning and end of the walking path. The three Kinect v2 sensors are highlighted with shaded boxes. (**B**): Holographic obstacle with one hurdle (height 0.4 m, depth 0.02 m). (**C**): Holographic obstacle with two hurdles (height 0.4 m, depth 0.3 m).

**Figure 2 sensors-20-01095-f002:**
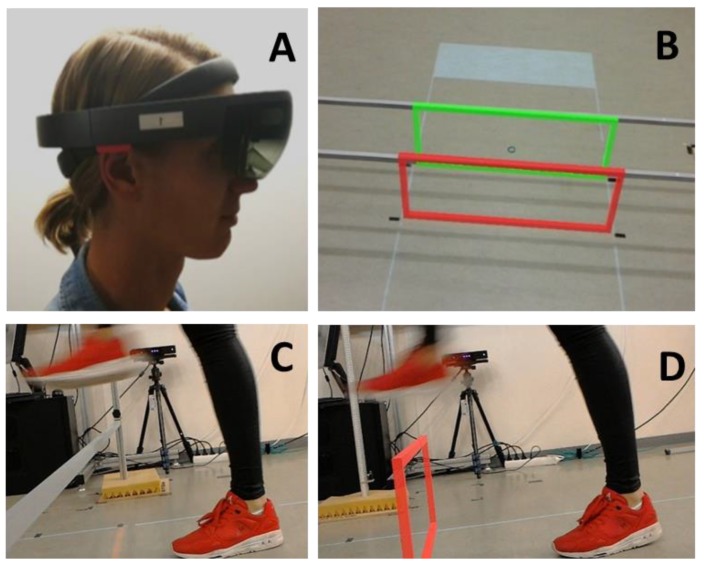
Person wearing a HoloLens mixed-reality headset (**A**), through which digital holographic obstacles can be merged with real physical obstacles of similar height and depth (**B**); all participants reported visual overlap). Examples of avoiding real (**C**) and holographic (**D**) obstacles (both 0.30 × 0.02 m).

**Figure 3 sensors-20-01095-f003:**
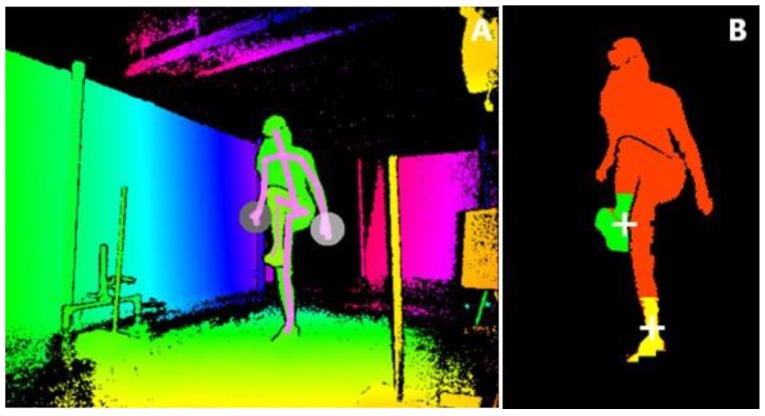
(**A**): Mirrored depth image from Kinect Studio with a superimposed stick figure based on the 25 body points of a participant avoiding a 3D holographic obstacle. (**B**): Extracted point clouds (mirrored) of left (yellow) and right (green) feet to determine step height and foot clearance with respect to the obstacle.

**Figure 4 sensors-20-01095-f004:**
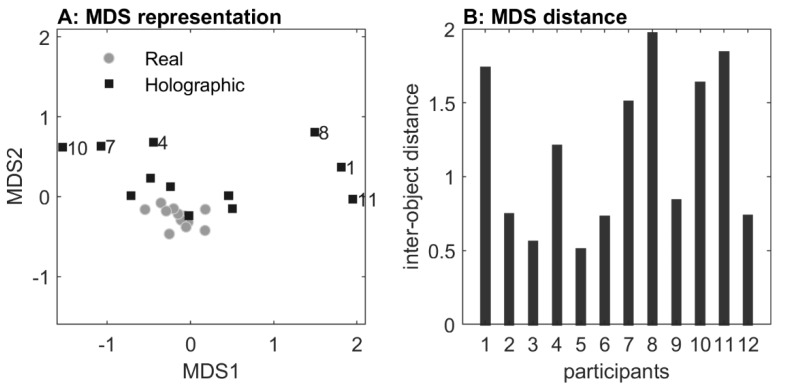
Multidimensional scaling (MDS) representation of obstacle-type conditions for each participant, derived from step heights and foot clearances of hurdle 1 and 2 for both feet (panel (**A)**). Each dot represents a participant, for real (gray) and holographic (black) obstacle conditions. Greater distances from the gray cluster of real obstacles is indicative of dissimilar holographic avoidance responses, as observed for participants 1, 8, and 11, and 4, 7, and 10 (cf. black squares labeled with participant number). Panel (**B**) shows the derived distance between each pair of real and holographic crossings for each participant.

**Figure 5 sensors-20-01095-f005:**
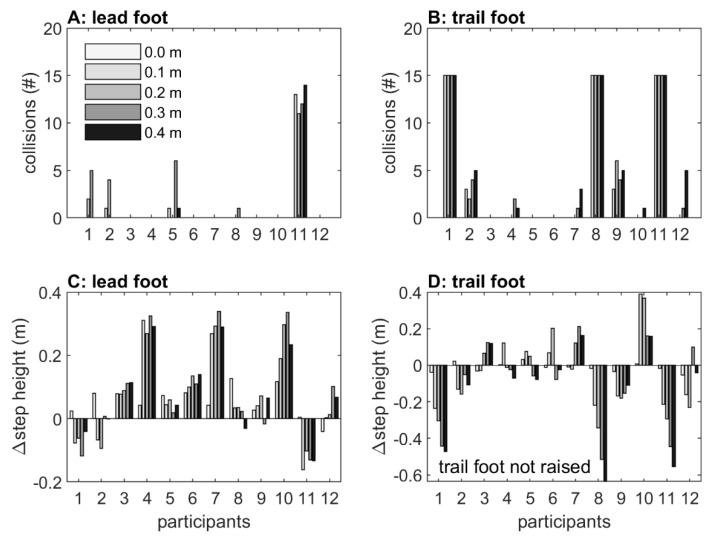
Number of collisions at different obstacle heights of hurdle 1 and 2 combined for every participant in mixed reality for the lead (panel **A**) and trail foot (panel **B**). The difference in maximum step heights between holographic and real obstacles at different obstacle heights at hurdle 1 for the lead (panel **C**) and trail foot (panel **D**).

**Figure 6 sensors-20-01095-f006:**
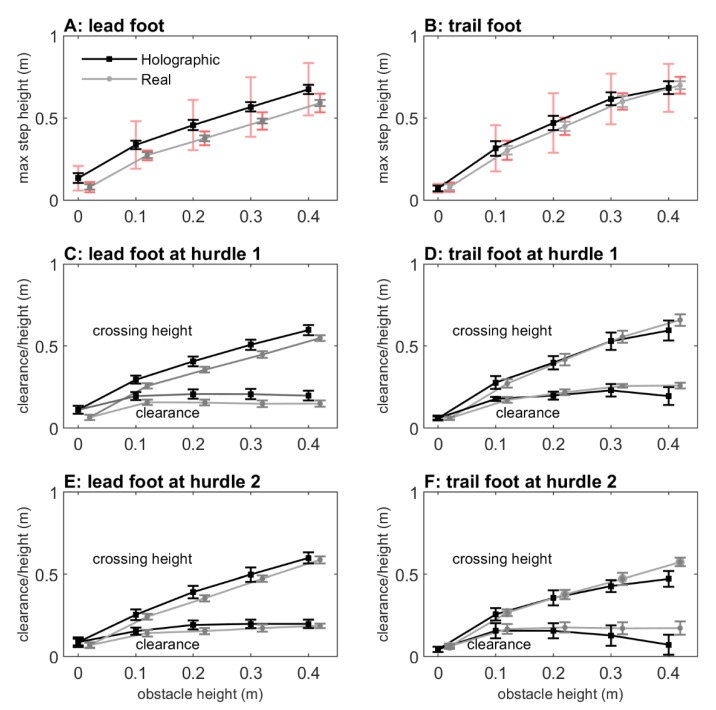
Maximum step height for lead (panel **A**) and trail foot (panel **B**), including indications of the within-subject (gray and black) and between-subject standard deviation (red; much greater for holographic than real obstacles). The crossing height and foot clearance at hurdle 1 for lead (panel **C**) and trail foot (panel **D**), including indications of within-subject variability (gray and black). Likewise at hurdle 2 (panels **E** and **F**). Note that participants 1, 8, and 11, who did not raise their trail foot for holographic obstacles, were excluded from panels B, D, and F.

**Figure 7 sensors-20-01095-f007:**
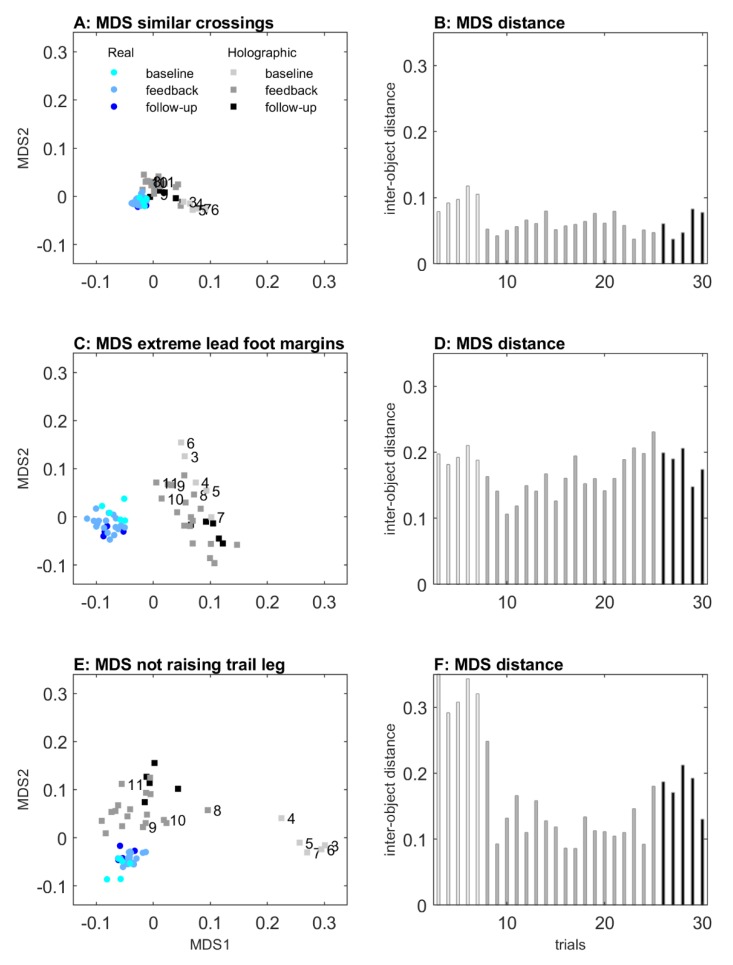
MDS representation of 2 blocks of trials 3 to 30 with real and holographic obstacles, derived from the foot clearances and maximum step heights of both feet from all 12 participants. Panel (**A**) shows participants identified in Experiment 1 as those with similar crossing maneuvers for real and holographic obstacles (participants 2, 3, 5, 6, 9, and 12). Panel (**C**) shows participants identified in Experiment 1 as those with exceptionally high lead-foot crossings (participants 4, 7, and 10), and panel (**E**) shows participants identified as those not raising their trail foot in mixed reality (participants 1, 8, and 11). Each trial object is represented by its trial number and obstacle type (real obstacles: blueish circles, holographic obstacles: grayish squares) and shaded as baseline (trials 3 to 7: brightest color), feedback (trials 8 to 25), and follow-up (trials 26 to 30: darkest color). Panels (**B**), (**D**), and (**F**) show the distances derived from the clearance and maximum step height of both feet, between individual holographic obstacle trials, and the average over all real obstacle trials, again with shades of gray indicating baseline, feedback, and follow-up blocks of trials.

**Figure 8 sensors-20-01095-f008:**
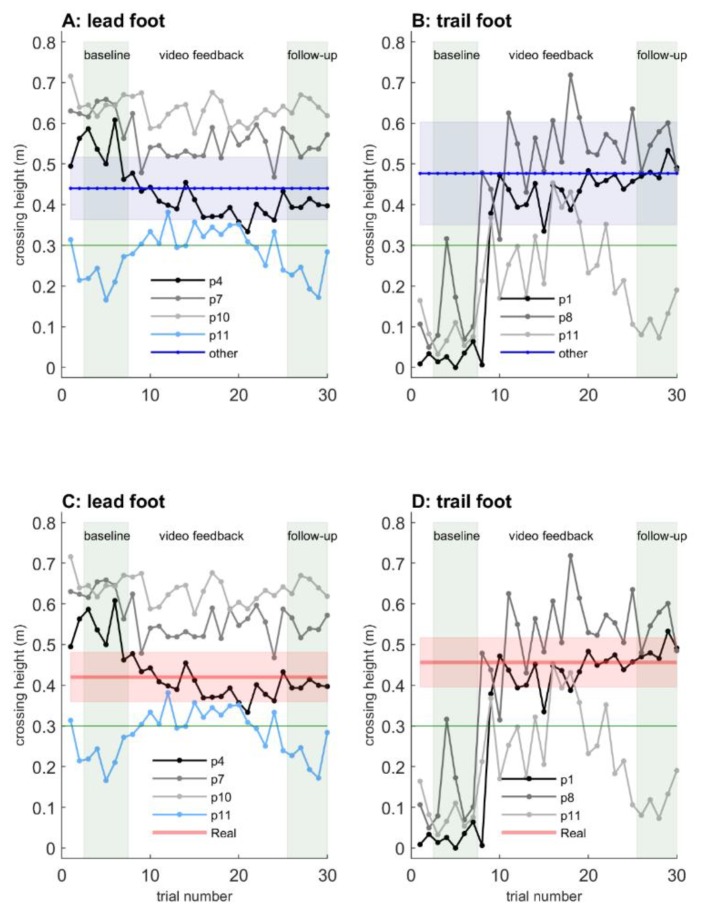
Crossing heights for lead (**A** and **C**) and trail (**B** and **D**) foot at holographic hurdle 1 (0.3 m height, marked as horizontal green line), focusing on the participants identified in Experiment 1 with dissimilar crossing maneuvers between real and holographic obstacles (Panels A and C: participants 4, 7, and 11 with extreme lead-foot margins; Panels B and D: participants 1, 8, and 11 without trail foot adjustments). The baseline and follow-up blocks of trials are marked by transparent gray boxes. Trial 8 is the first trial after the participant viewed mixed-reality video feedback. Trial 1 and 2 are familiarization trials. The blue transparent patches (Panels A and B) represent the 95% confidence interval of all trials of the nine participants with similar crossing maneuvers for real and holographic obstacles. The red transparent patches (in Panels C and D) represent the 95% confidence interval of all 30 real obstacle-avoidance trials of the three participants identified with dissimilar crossing maneuvers between real and holographic obstacles. Individual data falling outside these bands may be considered to deviate from, respectively, holographic obstacle crossing heights for participants with similar crossing maneuvers for real and holographic obstacles (blue band) and the real obstacle crossing heights for the participants identified with dissimilar crossing maneuvers between real and holographic obstacles (red band).

**Table 1 sensors-20-01095-t001:** Individual’s 95% confidence intervals of crossing heights of real and holographic obstacle-avoidance trials of Experiment 2 for participants identified in Experiment 1 as those with extreme lead-foot margins or without raising the trail foot.

**Participants Identified in Experiment 1 with Extreme Lead-Foot Margins**				
	Real obstacles	Holographic obstacles		
	All 30 trials	Baseline trials	Feedback trials ***	Follow-up trials
4	0.327–0.394 *	0.418–0.659	0.347–0.398 ** (9)	0.382–0.417 **
7	0.373–0.535 *	0.548–0.707	0.511–0.533 ** (5)	0.501–0.591
10	0.378–0.514 *	0.607–0.682	0.577–0.646 (14)	0.598–0.687
**Participants Identified in Experiment 1 without Raising Their Trail Foot**				
	Real obstacles	Holographic obstacles		
	All 30 trials	Baseline trials	Feedback trials	Follow-up trials
1	0.393–0.514 *	–0.020–0.076	0.275–0.400 ** (1)	0.434–0.542 **
8	0.420–0.503 *	–0.058–0.352	0.418–0.544 ** (1)	0.429–0.647 **
11	0.375–0.532 *	0.010–0.125	0.197–0.323 ** (1)	0.024–0.213

* Individual’s difference in crossing height between real obstacle avoidance trials (95% confidence intervals based on all 30 trials) and the baseline holographic obstacle-avoidance trials (95% confidence interval based on the five baseline trials). ** Individual’s difference in holographic obstacle crossing heights between the baseline trials and the feedback and/or follow-up trials. *** The number between parentheses indicates the first block of five consecutive feedback trials for which the 95% confidence interval deviated from that of the baseline trials (with 14 indicating the final block of five trials of mixed-reality video feedback).

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
