# Peer review of "Avoiding 3D Obstacles in Mixed Reality: Does It Differ from Negotiating Real Obstacles?"

_sensors, 2020, doi:10.3390/s20041095_

Round 1

Reviewer 1 Report

This manuscript presents an investigation into the HoloLens 1’s ability to provide locomotor interactions for (relatively) young, healthy participants. The study is well written and contains a systematic investigation of obstacle avoidance in real and holographic contexts. There are several concerns that arise when reading this manuscript and considering its suitability for publication in its current format.

First, the sample presented in the study, while not uncommon in studies of obstacle avoidance in young healthy participants, is too small and too heterogenous in the current case. The age range is large (a 64 years old participant, although not formally an “older” adult, may vary greatly from a 21 years old participant  in terms of obstacle avoidance strategies), and the results demonstrate a large heterogeneity in obstacle avoidance strategies. The authors divide the strategies (participants, really) into similar and “maladaptive” – where 50% of the sample belongs to one of 2 “maladaptive” strategies. This is a problematic division given the small sample size (if a “maladaptive” strategy is this common in an AR environment – is it really “maladaptive”? it may be that people select a different, equally valid, strategy when in an AR environment). I therefore believe that the sample in the study should be increased to be able to provide valuable information regarding the frequency of occurrence of these strategies. Furthermore, no information is provided on these “maladaptive” responders – e.g. age, cognitive ability, sensory function, etc. to explain why their responses were such – the discussion does note that the students were NOT among the maladaptive responders. This supports the hypothesis that this ability to interact with AR obstacles may vary with different factors, e.g. age or cognitive ability – but these data need to be at least presented. If, with a larger, and more heterogenous sample, the division remains similar, then I believe that the entire viewpoint of this paper should reflect the heterogeneity of individuals’ ability/choices when interacting with an AR environment during gait.

A second important point is that while it is clear (and the authors note this themselves) that AR technology is still in its infancy, the authors attribute the results of the current study largely to the inherent technological limitations of HoloLens. The question is, again, what part of this investigation alludes to the general nature of human interaction within an augmented reality setup. It is this part which needs to be emphasized, since already – better and more accurate AR systems are present (e.g. HoloLens 2 which may have increased the FOV significantly). Thus, the focus of the discussion should be equally on the human aspect as well as the technological one.

A third important point concerns statistical analysis and methodology – for experiment 1, the authors use an RM-ANOVA analysis with a very small sample of N=9 people (after excluding one group of “maladaptive” responders – why not the other group BTW?..) – it is not clear to me that the data would be normally distributed – such that this model may be safely used. In Experiment 2, the authors provide no statistical analysis and merely describe the different responses to feedback. This part is problematic and in fact, the one figure which is provided does not seem to support the claim that people modify their strategies given the feedback – nor maintain this change at follow-up. If these claims are made then they need to be supported statistically.

Finally, the 2nd experiment provides feedback in the form of an offline video in sagittal view of the movement. The rationale for selecting this type of feedback is unclear, and as the authors state themselves – it may be suboptimal. In the context of a study examining AR technology, using offline 2D video as feedback seems somewhat anachronistic. I also am not sure that this video provides knowledge of results, since it is unclear what the task was (did the “maladaptive” participants know that the trail leg was supposed to rise higher?)

Reviewer 2 Report

The authors presented a well-structured investigation on the use of HoloLens headset for obstacle negotiation research. The overall presentation format is solid, with proper experimental design. One major limitation is the small and heterogeneous sample size, leading to heterogeneous outcome measures (half participants classified as maladaptive responder), and a somewhat lengthy/convoluted data interpretation. None the less, the novelty of the technology and its potential still warrant a high impact publication. Some minor revisions are needed for better data interpretation as well as clarification on the experiment setup.

Experiment setup.

The walking path is 2.1m long, with the obstacle placed at 1.2 m distance away from the starting box, and 0.9 m from the stopping box. Given the distance, the obstacle crossing task happened during a transition phase of gait initiation, as well as gait termination. Whereas other literature often focused on obstacle negotiation during stable locomotion or as a discrete stepping task. It would be best the authors could discuss the reasoning of the walking path selection, and if the findings from this setup could translate to stable locomotion settings. 

Additionally, did the participant wear the HoloLens during the physical obstacle test? Even without the display, the visor may already limit the FOV of surroundings and affect the obstacle crossing strategy. The weight of the headset may also affect the stability control of balance/locomotion, thus affecting the crossing strategy. This could be a limitation of the current study.  A side note, did any participant wear glasses while using the HoloLens?

It has been also documented that the user comfort deteriorates after wearing the headset for an extended time (Cometti, C., Païzis, C., Casteleira, A., Pons, G., & Babault, N. (2018). Effects of mixed reality head-mounted glasses during 90 minutes of mental and manual tasks on cognitive and physiological functions. PeerJ6, e5847.). It would be nice if the authors can elaborate on how long the total test last, if user comfort was documented, and if it may affect the outcome measures.

Results/Discussions:

Experiment 1

Given the diverse range of participant’s age, and authors’ statement in discussion that none of the young students were identified as maladaptive responder, it is best to include individual’s age for those maladaptive responders (P1,8,11) & (P4,7,10), and to discuss if age is a key factor for the maladaptive behavior.

It is inconsistent in the results that (p1,8,11) were excluded for the trailing foot analysis, due to its bias, but (p4,7,10) were included even though it may bias the lead foot analysis.  Would it be possible to run stats analysis in three clusters and report it in a table for pairwise comparison?  

Discussion:

Line 379-382, the authors mentioned several factors that may affect the maladaptive response, it would be best if the authors can provide more information regarding the participant demographic/personal trait that may also contribute to the maladaptive response (i.e. age, gender, wearing glasses, occupation, tech-savviness, previous experience with VR/AR Headset, user comfort after use, etc.)

Experiment 2

It is understandable that as a pilot project, a second HoloLens is used to record the side view of the crossing (with 3D obstacle visualized in the view) to provide visual feedback on a computer screen after each trial. But since the location of the obstacle would not change during the experiment 2, wouldn’t a single side-view webcam with a virtual mark of the obstacle height provide the same information? And the side-view video can also be streamed to the HoloLens after each trial to provide instant feedback? A live alert for collision may also be generated based on the side view (if foot clipped with the virtual hurdle).

Regardless, the results section still hold that feedback can alter maladaptive response, with some retention effect. The results presentation (Fig 8) may need some editing, as the lines representing others (baseline and follow up) may overwhelm the readers. It may be best to generate a confidence Interval for the others.

It seems the author attributed the potential reason for p11’s behavior with fatigue, but did not fully explain why p9’s behavior. Did p9’s experiment 2 baseline fall below its experiment 1 values? Can it be attributed to fatigue as well?  Are any self-reported fatigue measures available? Are the fatigue causing by wearing the headset or doing the stepping repetitively? Mental or physical fatigue?  If the fatigue did not affect the physical obstacle crossing, then what could be the reason for the fatigue to affect virtual obstacle crossing? It would be best if the authors can add a discussion section providing the possible explanations (maybe in the general discussion).

Reviewer 3 Report

This is the first round of review of the manuscript “Avoiding 3D obstacles in mixed reality: Does it differ from negotiating real obstacles?”. The two studies described in this manuscript are well designed. The authors managed to use these two studies to address the challenges and potentialities of the application of mixed-reality technology in the area of gait modification and motor learning. The manuscript is well written, with a clear introduction and important details in experimental design. The authors made a thorough discussion regarding their findings. Please find the detailed comments below.

Page 3 Section 2.1.1  It seems that there is a lot of variation in age and height in the group of participants. Especially the author used blocks with similar heights to all participants without normalizing the results to participants’ body heights. I think the authors need to normalize their results to adjust the difference in body heights.

Page 3 Line 120-121 I’m not very clear what the authors meant by saying “The order of the physical and holographic obstacle-avoidance blocks was counterbalanced over participants”. Could you explain it here?

Page 4 Line 147-148 “All participants reported visual overlap …”. Does that mean a participant’ self report is the only standard you used to adjust the location of the projection?

Page 6 Line 203-206  “Three trials were excluded … because of missing values”. Are the excluded “three trials” also collected from participant 10? Please clarify it here.

Page 7 Section 2.2.1

In this section, the authors described the subgroup difference in MDS values and distance. However, to me, it was not very clear if there’s any physical meaning of MDS1 and MDS2. I guess that MDS1 represented the change in step heights, while MDS2 is related to change in foot clearance. Please state in the manuscript or in Figure 4A clearly. Moreover, whether a positive or a negative MDS value relates to a certain change of direction is not clear. Please clarify. When classifying the two groups of participants, did the authors use MDS distances = 1 as a cutoff value? Or is there any previous study that can be used as a reference here to support this value? Could the authors also provide step height and foot clearance values in two visual conditions? It sounds more straightforward to me.

Page 8 Line 254

First sentence: “a single collision”: May I know how a collision is defined in physical obstacle trials? Same paragraph “A total of 39…”. From your previous description, a collision in holographic trials is defined when foot clearance value is negative. Please make it clear if that’s your definition to define a collision. Please also state clearly if you have also used other parameters to define a collision.

Page 12 First paragraph of Section 3: Were the participants given any instructions during the training process? If yes, then what kind of instructions were provided to them?

Round 2

Reviewer 1 Report

I am satisfied with the modifications made to the manuscript.